# When can in-context learning generalize out of task distribution?

Chase Goddard [1]   Lindsay M. Smith [1]   Vudtiwat Ngampruetikorn [*2]   David J. Schwab [*3]

## Abstract

In-context learning (ICL) is a remarkable capability of pretrained transformers that allows models to generalize to unseen tasks after seeing only a few examples. We investigate empirically the conditions necessary on the pretraining distribution for ICL to emerge and generalize *out-of-distribution*. Previous work has focused on the number of distinct tasks necessary in the pretraining dataset. Here, we use a different notion of task diversity to study the emergence of ICL in transformers trained on linear functions. We find that as task diversity increases, transformers undergo a transition from a specialized solution, which exhibits ICL only within the pretraining task distribution, to a solution which generalizes out of distribution to the entire task space. We also investigate the nature of the solutions learned by the transformer on both sides of the transition, and observe similar transitions in nonlinear regression problems. We construct a phase diagram to characterize how our concept of task diversity interacts with the number of pretraining tasks. In addition, we explore how factors such as the depth of the model and the dimensionality of the regression problem influence the transition.

## 1. Introduction

The ability of transformers (Vaswani et al., 2017) to do few-shot learning from examples seen in their context is a striking phenomenon exhibited by modern large language models called *in-context learning* (ICL) (Brown et al., 2020). ICL has been extensively studied (Raventos et al., 2023; Lu et al., 2024; Garg et al., 2023; Chan et al., 2022; Singh et al., 2023) and enables models to solve certain new tasks

without re-training. Of particular interest is how the ability for transformers to perform ICL arises from pretraining and what the limits of ICL generalization are: What conditions must be met in order for ICL to emerge and generalize *outside* of the pretraining distribution?

Prior work (Raventos et al., 2023; Lu et al., 2024; He et al., 2024) has focused on understanding how the number of pretraining tasks affects the ability of the model to generalize to unseen tasks (generated from the same distribution as the pretraining tasks). Here, we ask a related but distinct question: If a model is pretrained only on tasks from a *subset* of the full task space, what conditions are necessary for it to generalize to the rest of the space? We think of this question as asking about the *out-of-distribution* (OOD) performance of models trained to do in-context learning. This question prompts us to consider a more general notion of *task diversity* that depends not only on task enumeration but also on how different or similar they are. Sampling a distribution with many similar tasks has the potential to induce the model towards a more specialized ICL solution that performs well only on novel tasks *within* its pretraining distribution. However, we observe that transformers trained to do ICL of linear functions undergo a transition from a specialized solution to one that generalizes over the full task space as we increase the degree of task diversity. This phenomenon of out-of-distribution *task generalization* sheds new light on in-context learning behavior[1].

### 1.1. Related work

Our investigation into the effects of task diversity on the emergence of ICL was motivated by the results of Raventos et al. (2023) on pretraining task diversity for linear regression tasks in-context, where they find that the number of pretraining tasks impacts the emergence of ICL *in-distribution*.

Here, we investigate in-context learning of linear functions in the *out-of-distribution* setting, and investigate the *domain generalization* performance of transformers. Our experimental setups are intentionally similar to those in Raventos et al. (2023) in order to aid comparison between in- and out-of-distribution ICL behavior. See e.g., Gulrajani & Lopez-Paz (2020); Arjovsky (2021); Liu et al. (2023); Yang et al. (2023)

---

[*]Equal contribution [1]Joseph Henry Laboratories of Physics, Princeton University, Princeton, NJ, USA [2]School of Physics, University of Sydney, Sydney, Australia [3]Initiative for the Theoretical Sciences, The Graduate Center, CUNY, New York, NY, USA. Correspondence to: Chase Goddard <cgoddard@princeton.edu>.

*Proceedings of the 42nd International Conference on Machine Learning*, Vancouver, Canada. PMLR 267, 2025. Copyright 2025 by the author(s).

---

[1]Code available at https://github.com/cwgoddard/OOD_ICL

for more background on domain generalization.

Similarly, He et al. (2024) look at the setting of modular arithmetic ICL tasks and analyze how the number of pretraining tasks affects generalization. Lu et al. (2024) provide an exactly solvable model of linear attention transformers for linear regression ICL tasks and empirically show agreement of their theory with traditional transformers. Other work has analyzed how transformers implement an internal gradient-descent algorithm to learn tasks in-context, shedding light on the mechanistic properties behind ICL (Akyürek et al., 2023; Ahn et al., 2023; Von Oswald et al., 2023). Fu et al. (2024a) argue that transformers must be implementing second-order optimization methods instead in order to solve ICL linear regression tasks. Chan et al. (2022); Edelman et al. (2024); Singh et al. (2023); Nguyen & Reddy (2024) investigate how properties of the data and optimization dynamics impact the emergence of ICL throughout the course of training.

### 1.2. Distribution Shift & Generalization

In-context learning is a powerful capability of language models, and in order to build trust in models, we should better understand how well such capabilities extend to novel tasks beyond those in the training data.[2] In addition to generalizing to tasks that interpolate between those seen in training, we'd like models that generalize to genuinely novel tasks. Whether this is possible depends on the nature of the distribution shift, and this question is in the purview of out-of-distribution generalization but at the level of tasks instead of samples.

In this work, we characterize when transformers develop in-context learning behaviors that are robust to task distribution shift. Since ICL can be thought of as generalization to novel tasks, here we ask when models succeed at *domain generalization* with respect to tasks. Indeed, a common assumption in the literature on out-of-distribution generalization (Gulrajani & Lopez-Paz, 2020; Arjovsky, 2021; Liu et al., 2023; Yang et al., 2023) is the so-called "covariate shift" assumption where the conditional distribution of the label given the input is held fixed, but the distribution of inputs changes at test time. In the context of tasks, we consider sampling tasks during training from a subset of the task space and ask about generalization to tasks outside of that subset. All tasks share the common property of being linear relationships, and thus we are studying domain generalization of tasks.

---

[2]We note that another perspective is to train on as large and diverse a pretraining dataset as possible so nothing is out of distribution, but there will always be novel tasks that have not been encountered before.

### 1.3. Contributions

Our core contributions are as follows:

- We train transformers to exhibit ICL of linear functions with weight vectors drawn from a subset of the unit hypersphere. As the size of this subset increases, we observe a transition from specialized models, which perform well only on the training portion of the hypersphere, to models that generalize out-of-task-distribution to the entire hypersphere.

- We investigate the nature of the solutions found by our transformers, and find that specialized solutions outperform optimal Bayesian solutions to the regression problem on small numbers of examples. In contrast, transformers that generalize to the entire hypersphere exhibit performance similar to optimal solutions.

- We examine how two notions of task diversity (number of tasks and task similarity[3]) interact, and construct a phase diagram that reveals three distinct regimes of ICL generalization.

- We show that specialization-generalization transitions also occur in nonlinear regression problems, suggesting that the phenomenon may be a general feature of ICL in transformers.

## 2. Training setup and task distribution geometry

**ICL of linear functions:** We investigate the ability of transformers to perform in-context learning of linear functions when pretraining tasks are drawn from distributions with varying levels of *task diversity*. Specifically, each *task* is a linear map in $d$ dimensions, $w \in \mathbb{R}^d$, and we control task diversity by sampling tasks from hyperspherical caps of varying half-angles. The transformer takes as input a sequence of up to $n$ pairs $\{x_1, y_1, \ldots, x_n\}$, where $y_i = w^T x_i + \epsilon_i$, with $x_i \sim \mathcal{N}(0, I_d)$ and $\epsilon_i \sim \mathcal{N}(0, \sigma^2)$.

**Pretraining task distribution:** We define a family of *task distributions* parameterized by $\phi \in [0, \pi]$ (See Fig 1B). We take $S^{d-1}(\phi)$ to be a section of the surface of the hypersphere in $d$ dimensions, i.e. $S^{d-1}(\phi) = \{w \in S^{d-1} \mid \text{angle}(w, v) \leq \phi\}$, with $v \in \mathbb{R}^d$ a fixed vector. We then define the task distribution as a uniform distribution on this spherical cap, i.e., $p_\phi(w) \equiv \text{Unif}(S^{d-1}(\phi))$. For details on how to effectively sample from such a distribution, see section A.5 in the appendix.

---

[3]For linear problems, it is natural to choose the inner product between tasks $w_1^T w_2$ as a similarity measure.

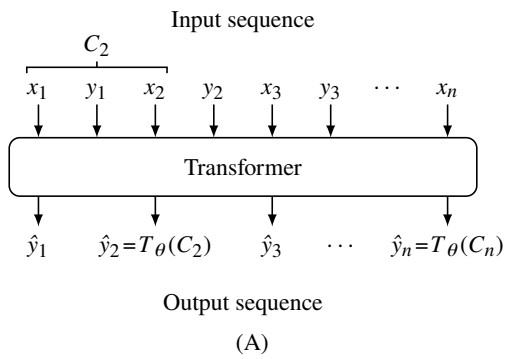

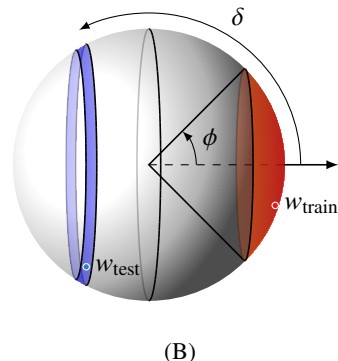

(A)                                                              (B)

Figure 1: **Testing ICL generalization via task similarity. A:** The transformer takes as input a sequence of pairs $\{x_i, y_i\}_{i=1}^n$ and is trained to predict $y_k$ from a context $C_k = \{x_1, y_1, \ldots, x_k\}$. The elements $x_i$ and $y_i$ are related linearly by a task $w$ via $y_i = w^T x_i + \epsilon_i$. **B:** The training tasks $w_{\text{train}}$ are drawn from a hyperspherical cap with half-angle $\phi$ (with $\phi = 180°$ corresponding to the entire hypersphere). The test tasks $w_{\text{test}}$ are drawn from a hyperspherical band of width $\Delta\delta$ starting an angle $\delta$ away from the "pole" of the sphere.

**Pretraining:** During pretraining, the transformer $T_\theta$ is optimized to minimize the mean squared error (MSE) between a *context* of data $C_k \equiv \{x_1, y_1, \ldots, x_k\}$ and the target $y_k$. During pretraining, the tasks $w$ are drawn i.i.d. for each context from $p_\phi(w)$. We use AdamW (Loshchilov & Hutter, 2017) to optimize the MSE,

$$L_{\text{train}}(\theta) = \mathbb{E}_{w \sim p_\phi} \left[ \frac{1}{n} \sum_{k=1}^n \left( T_\theta(C_k) - y_k \right)^2 \right]. \quad (1)$$

**Test task distribution:** We evaluate the performance of the transformer over a family of task distributions parameterized by $\delta, \Delta\delta \in [0, \pi]$ (See Fig 1B). We define the hyperspherical band starting at angle $\delta$ with width $\Delta\delta$ to be the set $B^{d-1}(\delta, \Delta\delta) = \{w \in S^{d-1} \mid \delta \leq \text{angle}(w, v) \leq \delta + \Delta\delta\}$, with $v$ some fixed vector. The test task distribution is then uniform over this set: $p_{\delta, \Delta\delta}(w) = \text{Unif}(B^{d-1}(\delta, \Delta\delta))$. For more on how to sample from these bands, see section A.5.2 in the appendix.

**Evaluation:** We evaluate models by computing the MSE between the full context $C_n$ and the final target $y_n$. During test time, we draw $w$ i.i.d. for each context from $p_{\delta, \Delta\delta}$:

$$L_{\text{test}}(\theta) = \mathbb{E}_{w \sim p_{\delta, \Delta\delta}} \left[ \left( T_\theta(C_n) - y_n \right)^2 \right] \quad (2)$$

## 3. Experimental Results

Unless stated otherwise, we study $d = 10$ dimensional regression with $n = 50$ examples in each context. We use a GPT-2 style transformer (Radford et al., 2019) with learned positional embeddings, a hidden dimension of $d_h = 128$,

10 layers (except in Fig 9), and 8 attention heads. We use a learned linear embedding to map $x_i$ and $y_i$ to the hidden dimension $d_h = 128$. The target values $y_i$ are padded with $d - 1$ zeroes. For further training details, see Appendix section A.1.

During pretraining, we train 12 models over pretraining distributions $p_\phi(w)$ for $\phi \in [15°, 180°]$ in $15°$ increments. We observe that repeated runs, with different initializations and trained on data generated from different sampled tasks $w \sim p_\phi$, yield consistent results[4] (see Fig 11). For small $\phi$, we also see the model go through two stages of specialization over the course of training (see Appendix A.2), the first of which is before this plateau.

### 3.1. Modes of out-of-distribution generalization

Before we investigate experimental results, it is instructive to discuss the ways in which models may (or may not) generalize out-of-task-distribution:

1. Models may fail to meaningfully generalize out-of-task-distribution. (For example, when prompted to solve a task $w_{\text{test}}$ not in the support of the pretraining distribution $S^{d-1}(\phi)$, models may simply pick the task $w_{\text{close}} \in S^{d-1}(\phi)$ that is closest to $w_{\text{test}}$, but fail to generalize beyond this level.)

2. Models may generalize out-of-task-distribution, but only achieve maximum performance when pretrained on the *entire* task space ($\phi = 180°$) .

---

[4]During training, the loss generically "plateaus," staying at a constant value for many steps before beginning to decrease. This phenomenon has been observed before, as in (Fu et al., 2024b).

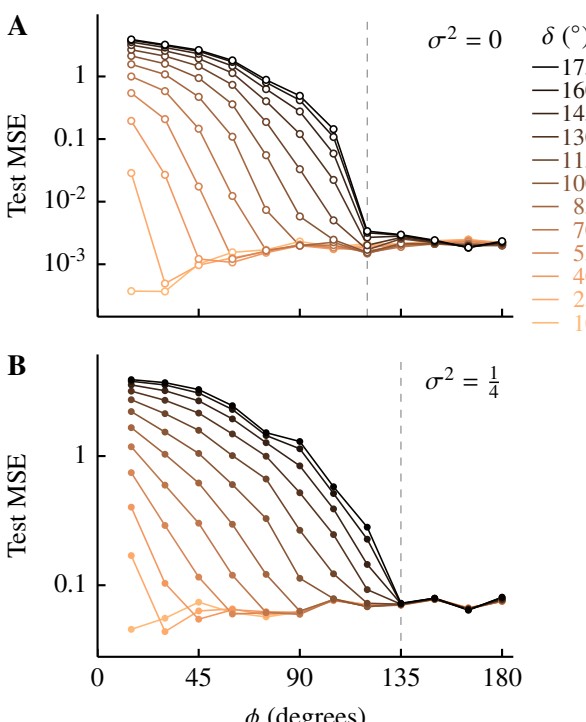

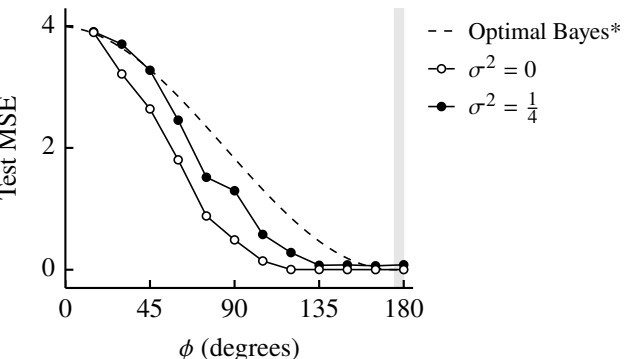

Figure 3: **Pretrained transformers outperform Bayes-optimal solutions in out-of-task-distribution generalization.** For $\delta = 175°$ (shaded grey region), we plot the excess test loss for models with varying pretraining distributions. The dashed line shows the test error for the optimal in-task-distribution Bayesian solution (see section 3.2).

Figure 2: **Task distribution diversity induces a transition from specialized to general-purpose ICL. A:** Test error in $\Delta\delta = 5°$ bands (see Fig 1) for transformers pretrained to do in-context learning of linear functions with pretraining task distributions $p_\phi(w)$. For distributions with $\phi \lesssim 120°$, the transformer learns a specialized solution that performs well on unseen tasks drawn from the $p_\phi(w)$, but fails for tasks outside this distribution. However, for pretraining distributions with $\phi \gtrsim 120°$, the transformer learns a solution that performs well for all test angles $\delta$. Here, the label noise is $\sigma^2 = 0$. **B:** With $\sigma^2 = 0.25$, we still observe a transition from a specialized to a generic solution, but the transition point has moved to $\phi \approx 135°$. The vertical axis measures the excess test error above the noise floor set by $\sigma^2$.

3. Models may generalize out-of-task-distribution in a *sharp* way with increasing $\phi$: there may be some $\phi_c < 180°$ such that models achieve maximum performance for all $\phi \geq \phi_c$.

We view the last option as perhaps the most striking: it implies that models can achieve optimal performance with incomplete data. In the following, we show that transformers can generalize out-of-task-distribution in this fashion.

### 3.2. Specialization-generalization transition

We show the results from evaluating these models on the test task distributions $p_{\delta,\Delta\delta}(w)$ in Fig 2. We pick $\Delta\delta =$

5° and examine a range of $\delta$ (see legend). For models with a pretraining task distribution with $\phi \lesssim 120°$, we observe good test performance only within the portion of the hypersphere covered by the pretraining distribution, and performance degrades outside of this range. However, for models trained over $p_\phi(w)$ with $\phi \gtrsim 120°$, we observe low test error for all $\delta$. In fact, the model performs similarly for all test angles $\delta$, suggesting that transformers learn a general-purpose solution in this regime. This occurs despite the fact that these models were trained using only data restricted to a *subset* of the full task space. Notice also that even before the transition, models trained on a cap with $\phi \geq 45°$ exhibit nontrivial out-of-distribution task generalization (Fig 3).

In Fig 2B, we see that the transition is sensitive to the level of label noise $\sigma^2$. For noisy regression with $\sigma^2 = \frac{1}{4}$, we see that the transition now occurs around $\phi \sim 135°$, but that the qualitative behavior of the transition is unchanged.

**Optimal Bayes & OOD generalization** How well do transformers generalize out-of-task-distribution? Here, we compare their out-of-task-distribution performance to that of the optimal Bayesian estimator for a given $p_\phi(w)$. Following (Raventos et al., 2023), we derive an expression for the optimal estimator of $y_n$ under the prior $p_\phi(w)$. Observe that in order to minimize $L_{\text{test}}(\theta)$, the optimal estimator is given by the posterior mean of $y_n$ conditioned on the full

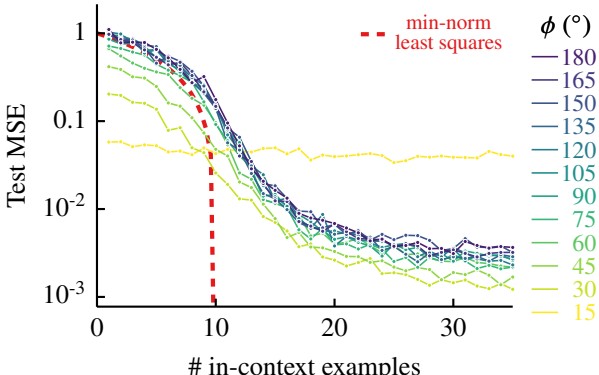

Figure 4: **Specialized ICL outperforms OLS for small context length.** We evaluate the models in-task-distribution for varying context lengths, and plot the performance of the transformer (solid) and ordinary least squares (dashed) for the same data. For low context length, the specialized solution learned by models with $\phi \lesssim 90°$ outperforms OLS. For $\phi = 15°$, the specialized solution is worse than OLS for large context length.

context $C_n$,

$$\mathbb{E}[y_n|C_n] = \int dw\, dy_n\, y_n p(w, y_n|C_n)$$
$$= \int dw\, dy_n\, y_n p(y_n|x_n, w)p(w|C_{n-1}, y_{n-1})$$
$$= \int dw\, w^T x_n p(w|C_{n-1}, y_{n-1})$$
$$\equiv \hat{w}^T x_n$$

with

$$\hat{w} = \frac{\int dw\, p(w)w \prod_{k=1}^{n-1} p(y_k|x_k, w)}{\int dw\, p(w) \prod_{k=1}^{n-1} p(y_k|x_k, w)} \qquad (3)$$

We now consider the choice $p(w) = p_\phi(w)$. Although the integrals in Eqn 3 are intractable in this case, it is clear that because $p_\phi(w)$ lacks support outside of $S^{d-1}(\phi)$, $\hat{w}$ must always be contained in $S^{d-1}(\phi)$. The optimal Bayesian estimator ($w_{\mathrm{OB}}$) for this problem therefore fails to meaningfully generalize out-of-task-distribution, as the best one can hope for is to assign $\hat{w}$ to that vector $w_{\mathrm{OB}} \in S^{d-1}(\phi)$ that is closest to the target task.

In Fig 3, the test loss for $w_{\mathrm{OB}}$ is given by the dashed line. Notice that the ability of pretrained transformers to outperform this optimal estimator out-of-task-distribution is therefore a direct consequence of the failure of these transformers to fit the optimal Bayesian solution. It remains an interesting avenue for further work to ask what form the prior $p(w)$ should take in order to give rise to the out-of-task-distribution performance we see here.

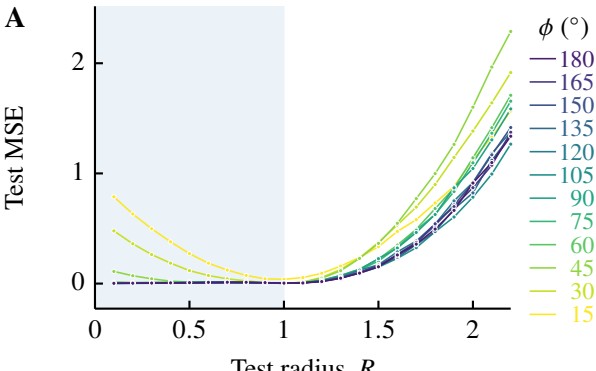

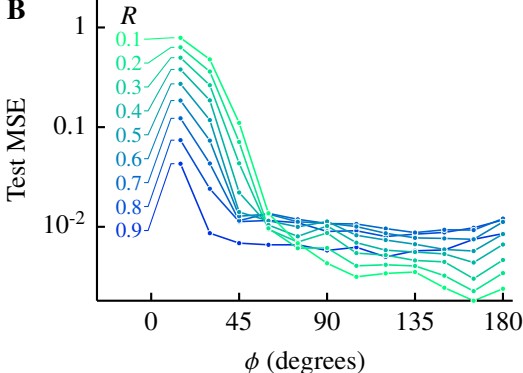

Figure 5: **Transformers trained to do ICL on the sphere generalize beyond it. A:** The test error for tasks drawn uniformly from subsets of a hypersphere of radius $R$, when a model is pretrained on tasks taken only from subsets of the *unit* hypersphere. When $\phi \gtrsim 45°$, the model generalizes to tasks with $R < 1$ (shaded), despite being pretrained with $R = 1$. **B:** Increasing task diversity drives generalization beyond the sphere: With sufficient task diversity ($\phi \gtrsim 45°$), transformers generalize not only to OOD tasks *on the sphere* (Fig 2), but also to OOD tasks *within* it.

**Comparison to ordinary least squares** We next investigate the solutions learned by the transformer on both sides of the transition. In Fig 4, we compare the performance of the transformer and ordinary least squares (OLS), solid and dashed curves, respectively. For short contexts, the specialized solution which is learned for $\phi \lesssim 120°$ outperforms OLS within the task distribution. For $\phi \gtrsim 120°$, the performance of the transformer is similar to OLS, except that the models' test error is not identically zero after $d = 10$ examples, unlike the least-squares solution. This sheds light on the nature of the specialization occurring: by fitting a strong prior to the pretraining data, models with low $\phi$ sacrifice out-of-distribution performance, but this bias enables them to outperform the general-purpose solution (OLS) for low context length. See also Appendix Figs 16 & 17 for models' performance for varying context length in other settings.

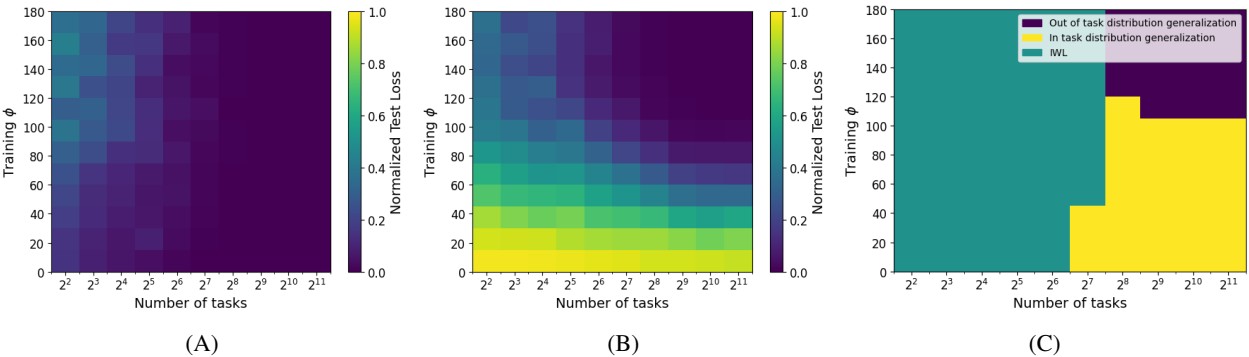

Figure 6: **Phase diagrams reveal three phases of generalization. A:** Phase diagram for in-task-distribution test loss ($\delta = 0°$). **B:** Phase diagram for out-of-task-distribution test loss ($\delta = 175°$). Diagonal structure reveals interplay between two the measures of task diversity (Training $\phi$, number of tasks $N$). **C:** Combining the earlier two phase diagrams reveals three phases of in-context learning. *In-weights learning, teal:* the model fits the training data but fails to generalize, both in- and out-of-task-distribution. *In-task-distribution generalization, yellow:* The model generalizes within the support of the pretraining distribution, but fails to generalize out of distribution. *Out-of-task-distribution generalization, purple:* The model generalizes well both in- and out-of-task-distribution. In constructing the phase diagram, we set the threshold between high and low generalization losses to $10^{-2}$.

**Beyond the unit hypersphere** What happens to the generalization ability of the model as the radius of the task distribution changes? We train several models on data generated from tasks on the surface of the unit hypersphere, and evaluate them on tasks drawn from spheres of varying radii. Each model is trained on tasks from $p_\phi(w)$ and evaluated on the equivalent distribution (with the same $\phi$) on a hypersphere with a different radius. In Fig 5, we observe that for $\phi \gtrsim 45°$ the model is able to generalize perfectly to tasks with $R < 1$ (shaded region), despite being trained only on tasks with $R = 1$. Increasing task diversity therefore drives the model to generalize not only to new portions of the hypersphere, but also beyond the hypersphere entirely.

### 3.3. Interplay between the two forms of task diversity

In order to examine the effect of both forms of task diversity (number of tasks and task similarity), we train 4 sets of 120 models (480 models total) with task similarity $\phi$ and number of tasks $N$ in the set: $(\phi, N) \in \{15°, 30°, \ldots, 180°\} \times \{2^2, 2^3, \ldots, 2^{11}\}$. In Fig 6A, we plot the resulting *in-task-distribution* loss averaged over the 4 sets: the loss for a test angle $\delta$ between $0°$ and $5°$ (these test angles are *always* in the training task distribution). In order to more effectively compare across different $\phi$, we normalize the loss for each $\phi$ by the maximum possible loss achievable from a predictor $\hat{w}$ and target $w^\star$ in $S^{d-1}(\phi)$. Without such a normalization scheme, models with small $\phi$ would trivially outperform those with larger $\phi$, since even an *arbitrary* choice of estimator $\hat{w} \in S^{d-1}$ cannot be too far from the target $w^\star$. We see that models with low $N$

and large $\phi$ perform poorly in-distribution, suggesting that the density of tasks may be important. For a more detailed analysis of the in-distribution performance of these models, see Appendix A.4.

In Fig 6B, we plot the resulting *out-of-task-distribution* loss averaged over the 4 sets, corresponding to test angles $\delta = 175°$. In order to compare effectively with the normalized in-task-distribution results, we normalize these losses by dividing by 4, the maximum loss possible when $\hat{w}, w^\star$ are on the sphere[5]. We see that models with small $\phi$ perform poorly, and observe a diagonal boundary dividing models that generalize well and those that do not, suggesting interplay between these two forms of task diversity. In 6C, we summarize these results as a phase diagram, depicting three distinct phases with a threshold of 0.01 as a cut-off between low loss (good generalization) and high loss (poor generalization):

1. Good generalization both in- and out-of-task-distribution (top right, purple).

2. Good in-task-distribution generalization, poor out-of-task-distribution generalization (bottom right, yellow).

3. Poor generalization both in- and out-of-task-distribution (left, teal); the model exhibits only in-weights learning (IWL).

**Scaling Dimension and Depth** In Fig 8, we investigate how the transition changes with changes in $d$, the dimen-

---
[5]To see this, set $\hat{w} = -w^\star$.

sionality of the regression problem. Unlike what is observed in Raventos et al. (2023), where as $d$ increases, a greater pre-training task diversity $N$ is necessary to induce ICL, here we observe that as $d$ increases the transition along the $\phi$ axis does not seem to change location. This suggests that the transition along this axis is not merely a result of high-dimensional geometry – in low dimensions (e.g., $d = 3$), the transition is still around $\phi \sim 120°$.

Similarly, we vary the number of layers in the transformer in Fig 9 to study how the transition from specialization to generalization changes with respect to model depth. We find that depth does not affect the angle ($\phi \sim 120°$) of the transition for out-of-task-distribution test loss ($\delta = 175°$). These models were trained using $N = 2^{11}$ pretraining tasks. We further show in Fig 19 in the appendix that the transition point remains the same across depth even for different testing angles $\delta$. These results further corroborate the phase diagram results in Fig 6 and show that model depth does not affect the transition point.

### 3.4. Classification

To investigate the generality of specialization-generalization transitions, we investigate the possibility of seeing a transition in a classification task: logistic regression. We now take the mapping between $x$ and $y$ to be:

$$y_i = H_{\frac{1}{2}}(\sigma(w^T x_i)) \tag{4}$$

where $H_{\frac{1}{2}}(\cdot)$ is the Heaviside step function with threshold $\frac{1}{2}$ and $\sigma(\cdot)$ is the logistic function. In Fig 7, we observe a specialization-generalization transition for this task. The transition now occurs at $\phi \approx 135°$. The fact that we observe a specialization-generalization transition in a classification setting hints that such transitions may be a more universal phenomenon.

### 3.5. Nonlinear regression

We now change the mapping between input and label for the regression to be a nonlinear function of the weights. Specifically, we consider $y_i = w_2^T \mathrm{ReLU}(W_1 x_i)$, with $x_i, w_2 \in \mathbb{R}^d$ and $W_1 \in \mathbb{R}^{d \times d}$. We choose $d = 3$ so that the model has 12 parameters. In Fig 10, we see that specialization-generalization transitions still occur, and investigate two ways of choosing the parameters. In Fig 10A, we pick the full 12-dimensional parameter vector $\theta = \{\mathrm{vec}(W_1), w_2\}$ from the surface of $S^{11}$. This choice induces a bias towards $\|w_2\| \ll 1$ for angles $\phi$ near the 'poles' ($v = (\pm 1, \vec{0})^T$). This bias is relaxed, however, when $\phi \sim 90°$, near the equator of the sphere. This leads to nonmonotonic behavior as $\delta$ changes – the tasks near the poles are more similar to each other than to those near the equator. In Fig 10A, we only show $\delta < 90°$ for this reason. In Fig 18 in the appendix, we show results for all $\delta$, and the

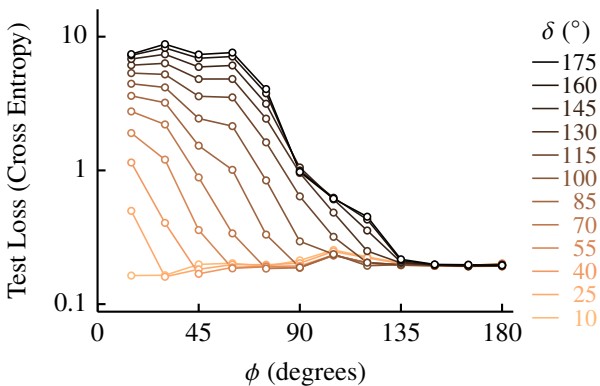

Figure 7: **Task distribution diversity induces a transition from specialized to general-purpose ICL in *classification* tasks:** We now consider logistic regression. $y_i = H_{\frac{1}{2}}(\sigma(w^T x_i))$. We see a similar transition to the one observed in the case of regression, but the transition occurs at $\phi \approx 135°$.

non-monotonicity can be observed where the red and blue lines, corresponding to $\delta$ near the poles, have lower test loss than the yellow lines, corresponding to $\delta \sim 90°$.

In contrast, in Fig 10B, we pick from two separate hyperspheres: $\mathrm{vec}(W_1) \in S^8$ and $w_2 \in S^2$. This choice leads to a qualitatively similar transition to those we see in the linear case, with a transition around $\phi \sim 135°$. Observation of a specialization-generalization transition beyond the linear regime suggests that such transitions may be a general phenomenon in ICL.

## 4. Discussion and future work

We propose another "axis" to training task diversity, distinct from the task diversity measure in (Raventos et al., 2023) (i.e., the number of pretraining tasks). This new axis of task diversity, based on the size of the subset of the task space, accounts for the similarity present between tasks. Depending on the level of task diversity present during pretraining, we have shown that transformers learn either a specialized solution that fails to generalize out-of-task-distribution, or a generic solution with good performance across the entire task space.

While we focus on the case of learning linear tasks, the phenomena of specialization-generalization transitions are likely more general. In particular, such transitions appear also in the presence of label noise (Fig 2) and in nonlinear regression problems (Fig 10). To extend our analysis to more complex tasks, it would be interesting to investigate ICL performance in richer settings as more general notions of "task similarity" are varied. In the context of linear

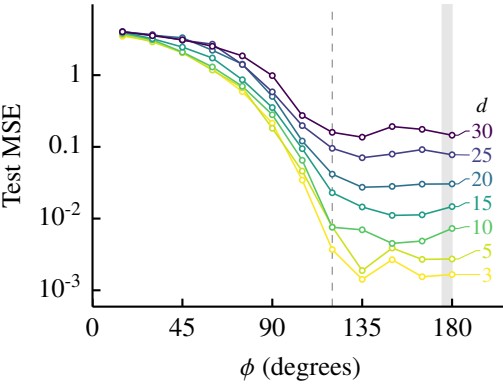

Figure 8: **Distributional diversity threshold is unaffected by task dimension.** Out-of-task-distribution test loss vs training spherical cap polar angle $\phi$ at various task dimensions $d$ (see legend). The test tasks are drawn from the spherical cap opposite to the "pole" of the training task distribution, $\delta = 175°$ (shaded region). We see that the threshold for out-of-task-distribution generalization stays close to $\phi \sim 120°$ (dashed line), regardless of the task dimension.

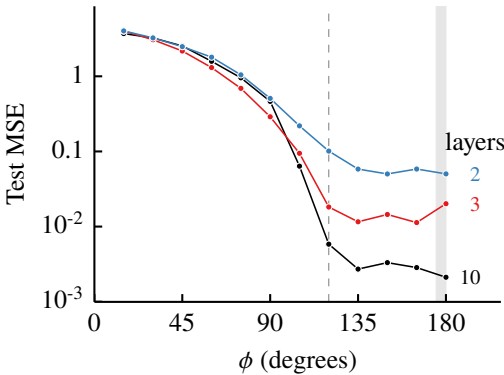

Figure 9: **Distributional diversity threshold is unaffected by model depth.** Out-of-task-distribution test loss vs training spherical cap polar angle $\phi$ from transformer models with two (blue), three (red), and ten (black) layers. The test tasks are drawn from the spherical cap opposite to the training task distribution, $\delta = 175°$ (shaded region). We see that the threshold for out-of-task-distribution generalization stays close to $\phi \sim 120°$ (dashed line), regardless of the model depth.

problems on the sphere, the similarity between tasks $w_1$ and $w_2$ is naturally measured by their inner product $w_1^T w_2$, but for more general problem settings it is less clear how to measure the similarity between tasks. Even within the linear setting, it may be interesting to explore task geometries beyond the sphere.

### 4.1. Types of distribution shift

In this work, we considered how transformers' ICL performance responds to a domain shift in the task distribution. To make this precise, we can formulate ICL as a supervised learning problem. Let's just consider the linear regression setting for now. Our supervised learning dataset is then:

$$\mathcal{D} = \{C_n, y_n\} \tag{5}$$

with $C_n = \{x_1, y_1, \ldots, x_n\}$, and $y_i = w^T x_i$. The problem of task generalization we consider is then a special case of the domain generalization problem. To see this, notice that during training we see a given context $C_n$ according to a distribution $C_n \sim p(w) \prod_{i=1}^{n} p(x_i)$. The label function $p(y_n|C_n)$ is deterministic given the context:

$$p'(y_n|C_n) = \delta(y_n - w^T x_n) \tag{6}$$

When we ask about contexts with $w$ outside the support of the training distribution, what we have done is to change the

data generating process to:

$$p(C_n, y_n) = p'(C_n)p(y_n|C_n) \tag{7}$$

$$= \left[ p'(w) \prod_{i=1}^{n} p(x_i) \right] \delta(y_n - w^T x_n) \tag{8}$$

Notice that this is a special type of covariate shift: we do not allow the full distribution $p(C_n)$ to vary – rather, we impose that 1) the sample generation distribution $p(x_i)$ remains unchanged; and that 2) we still draw $w$ independently from the samples $x_i$. It may be the case that this additional structure allows the network to more easily deal with the distribution shift.

This is not the only type of distribution shift one may care about, however, and prior work has investigated ICL under different distribution shifts. Ahuja & Lopez-Paz (2023) investigate instead a domain shift of the $x_i$'s, (i.e. $p(x_i) \to p'(x_i)$ and find that transformers fail to generalize when exposed to such shifts. We suggest that this may be because the authors do not expose the model to sufficient *data* diversity during pretraining. In Appendix A.6 we investigate the effect of data diversity in the linear regression setting and see a (noisy) transition in the behavior of the model's OOD generalization. Yadlowsky et al. (2024; 2023) instead consider ICL under *concept shift*: the mapping of data to labels $p(y_n|C_n)$ changes between train and test. The authors here find that transformers largely fail to generalize to such distribution shifts, even when exposed to a diversity of tasks during pretraining. Hill et al. (2025) consider

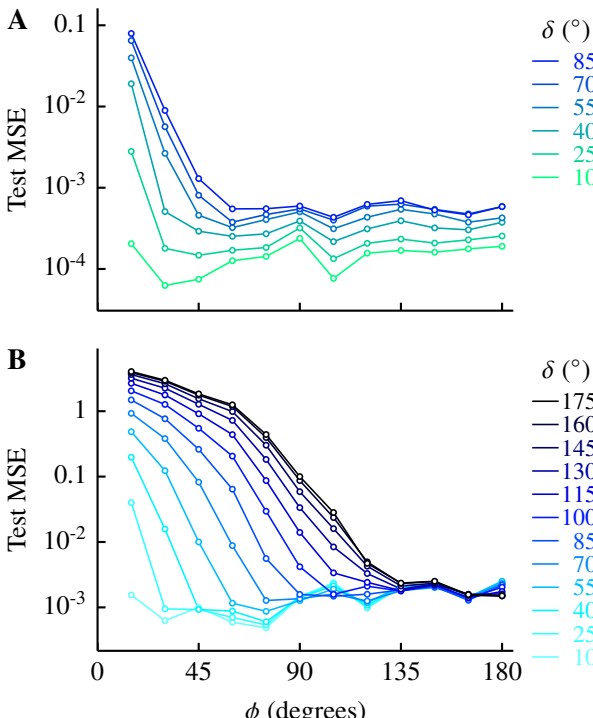

Figure 10: **Task distribution diversity induces a transition from specialized to general-purpose ICL in *nonlinear* regression tasks. A:** All parameters in the nonlinear model (a small one-hidden-layer network) are drawn from the same hypersphere. The transition occurs at $\phi \approx 60°$. Although the curves for each $\delta$ do not collapse onto each other after the transition, they remain within a band of size $\sim 10^{-3}$. **B:** The parameters in the nonlinear model are drawn separately from a different hypersphere for each layer in the model. The transition occurs at $\phi \approx 135°$.

ICL of linear functions where task vectors at test time are orthogonal to those seen during training, and again see that transformers fail to generalize to this distribution shift. We expect that this is because the natural task similarity measure for linear tasks, the inner product, is zero for orthogonal tasks. The ability our models display in successfully adapting to distribution shift between train and test time may therefore depend on the type of distribution shift considered, in addition to task/data diversity thresholds.

### 4.2. Future directions

A natural next step would be to develop an analytic theory for analyzing specialization-generalization transitions in transformers, similar to the analysis Lu et al. (2024) performed for the setting studied in Raventos et al. (2023). Such a model could provide further insight into the conditions necessary for out-of-task-distribution ICL to emerge.

Additionally, it would be interesting to study the dynamics of how out-of-task-distribution ICL emerges during training, shedding light on the learning processes & implicit biases of the optimization algorithms that enable models to generalize beyond their training task distribution.

Our experiments open a new direction for understanding how general-purpose models are able to solve unseen tasks using only a few examples in their context: we show empirically that transformers can learn to do ICL over much more of the task space than they are trained on. Understanding the generality of this behavior may help explain why language models are able to perform well on ICL tasks not present in their pretraining distribution. Although our experiments here are limited by their focus on relatively simple functions as the ICL task, we believe investigations into specialization-generalization transitions for more complex tasks are a promising direction for future study. Building trust in LLMs is an important challenge with positive societal impacts, and understanding the degree and nature of task generalization via ICL takes a step towards this goal.

### Impact statement

We present work on linear regression ICL tasks with the goal of understanding more broadly the factors that lead to the development of ICL behavior. While we focus on simplified experimental settings compared to the more complex ICL phenomena seen in LLMs, the impact of understanding the underpinnings of specialization and generalization in ICL will provide insights into the learning dynamics of LLMs, building trust in AI and producing positive consequences for society.

### Acknowledgments

CG acknowledges travel support from the National Science Foundation and by DoD OUSD (R&E) under Co-operative Agreement PHY-2229929 (The NSF AI Institute for Artificial and Natural Intelligence). LMS is supported by the National Science Foundation Graduate Research Fellowship Program under Grant No. DGE-2039656. VN acknowledges research funds from the University of Sydney. DJS was partially supported by a Simons Fellowship in the MMLS, a Sloan Fellowship, and the National Science Foundation, through the Center for the Physics of Biological Function (PHY-1734030).

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

# A. Appendix / supplemental material

## A.1. Further experimental details

All code was written in Python using the PyTorch library (Paszke et al., 2019). All models were trained for 58,000 steps using a batch size of 128 and a constant learning rate of $3 \times 10^{-4}$. All models were converged at the end of training. All models were trained on a single GPU, either a MIG GPU with 10GB of memory or an A100 with 40GB of memory, and took $\sim$ 3hrs to train.

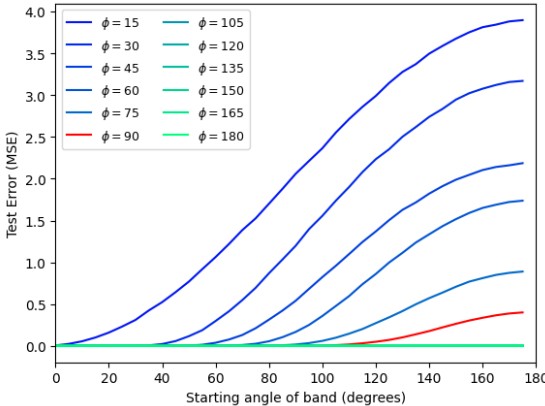

Figure 11: **A second run of Fig 2, with a different initialization and sampling of** $w \sim p_\phi$.

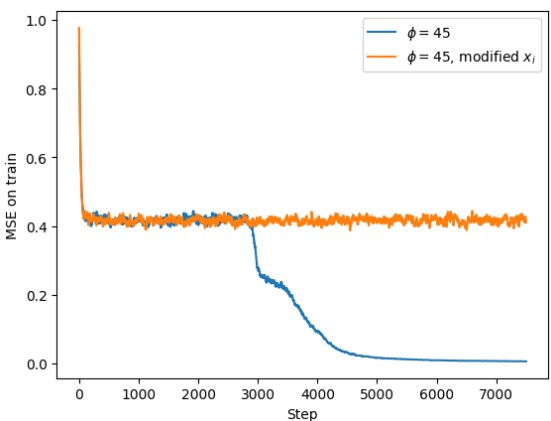

Figure 12: **Transformers undergo two stages of specialization during training: A:** For small $\phi$, the transformer rapidly (within the first epoch) learns a solution that only takes into account the component of $x_i$ in the direction of the vector $v$ forming the center of the hyperspherical cap. *Blue:* A transformer trained normally on training data with $\phi = 45°$. *Orange:* A transformer trained on data with the components of $x_i$ perpendicular to $v$ zeroed out. The training loss is smoothed with an exponential moving average for clarity of visualization.

## A.2. Two stages of specialization during training

In Fig 12, we compare the training loss of a transformer trained normally on data with $\phi = 45°$ to a transformer trained on modified data. To modify the data, we zero out all components of $x_i$ that are perpendicular to the vector $v$ defining the center of the hyperspherical training cap. We see that during early stages of training, the transformer trained on unmodified data performs similarly to the transformer trained on modified data, suggesting that early in training, transformers trained to do linear regression only take into account the component of $x_i$ parallel to $v$. Later in training, the unmodified transformer learns to take into account other directions in the training data. This suggests that there are two distinct specialized solutions learned by transformers when $\phi$ is small, the first of which is transient and disappears after training long enough.

## A.3. Defining the transition point

To more precisely define the transition point, we quantify the degree to which a model trained with a given $\phi$ performs similarly across test angles $\delta$. The intuition for this definition is that for a model with $\phi$ above the transition point, we should expect similar performance across all $\delta$ due to the rotational symmetry in the linear regression problem. To quantify this, we measure the standard deviation of the model's performance across $\delta$, and normalize this by the mean performance across $\delta$ :

$$\text{NSR} = \frac{\sqrt{\text{Var}_\delta \left[ \mathbb{E}_{w \sim p_{\delta, \Delta\delta}} [(T_\theta(C_n) - y_n)^2] \right]}}{\mathbb{E}_\delta \left[ \mathbb{E}_{w \sim p_{\delta, \Delta\delta}} [(T_\theta(C_n) - y_n)^2] \right]} \quad (9)$$

This definition resembles an inverse signal-to-noise ratio, or "NSR". To identify the transition point, we then look for a sharp drop in the NSR. In Fig 13, we plot the NSR against $\phi$ for the models in Fig 2A. We see a sharp drop in the NSR on a logarithmic scale, and identify which phase of learning we are in by thresholding the NSR. The fact that the drop is sharp means that our phase identification is not very sensitive to our choice of threshold.

## A.4. Comparison with dMMSE estimator

Plugging the uniform distribution over a finite pretraining set $\mathcal{W} = \{w_1, w_2 \ldots, w_N\}$ into Eqn 3 yields the discrete minimum mean-squared error (dMMSE) estimator (for the case of zero label noise):

$$\hat{w}_{\text{dMMSE}} = \arg \min_{w \in \mathcal{W}} \sum_{i=1}^{n} \left( w^T x_i - y_i \right)^2 \quad (10)$$

In Fig 14A, we investigate the in-distribution performance of pretrained transformers by comparing their performance to a dMMSE estimator. To do this, we interpolate between

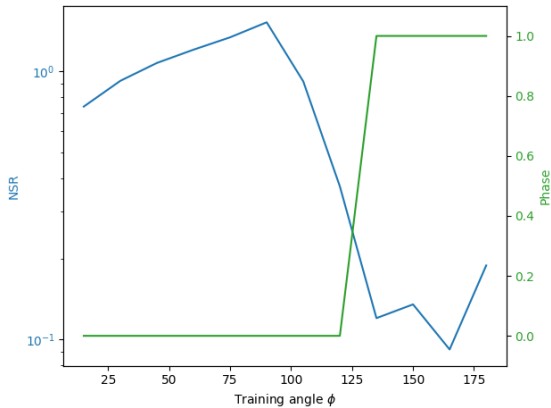

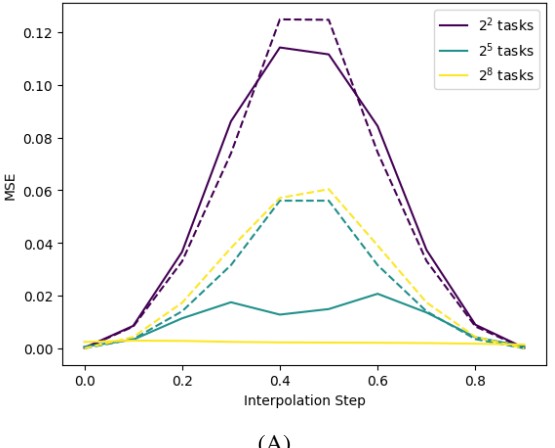

(A)

Figure 13: *Blue:* NSR (Eqn 9) vs $\phi$ for the models in Fig 2A. *Green:* Phase identification by thresholding the NSR, with a threshold of 0.5.

tasks in $\mathcal{W}$ along a great circle, and test both the transformer and dMMSE along this path. For a low number of tasks, transformers perform about as well as dMMSE, but transformers outperform dMMSE for larger numbers of tasks.

In Fig 14B, we plot a heatmap of the quantity:

$$D = \max_{\alpha} \left[ \left( \hat{w}_{\text{dMMSE}}^T x_\alpha - y_\alpha \right)^2 - \left( T_\theta(x_\alpha) - y_\alpha \right)^2 \right] \quad (11)$$

where $x_\alpha, y_\alpha$ is the regression problem generated by the interpolant weight vector $w_\alpha$ at interpolation step $\alpha$. The quantity $D$ measures the performance of the transformer relative to the dMMSE estimator: when $D < 0$, the transformer outperforms dMMSE. In the heatmap, we see that increasing number of tasks drives the transformer to a solution that outperforms dMMSE for all $\phi$. In order to compare across different values of $\phi$, we normalize $D$ using the scheme described in section 3.3. Notice that this normalization aligns the transition points for the transition from IWL to ICL across increasing number of tasks.

### A.5. Sampling from a portion of the hypersphere

The curse of dimensionality precludes sampling from a portion of the hypersphere via rejection sampling. Instead, we consider the problem of sampling uniformly from the intersection of the sphere with a cone in $d{+}1$ dimensions: i.e. sampling from the sphere $S^d(R)$ subject to the restriction:

$$\|w - v\|^2 \geq r^2 \Leftrightarrow \text{angle}(w, v) \leq \theta \Leftrightarrow w^T v \geq R^2 - r^2/2 \quad (12)$$

where $w$ is our sampled vector and $v$ is a fixed vector that defines the hypercone. The opening half-angle of the cone $\theta$ is related to $r$ via $r^2 = 2R^2(1 - \cos\theta)$.

WLOG, we take $v = R\hat{e}_1 = (R \ \vec{0})$, where $\vec{0} \in \mathbb{R}^d$. Then,

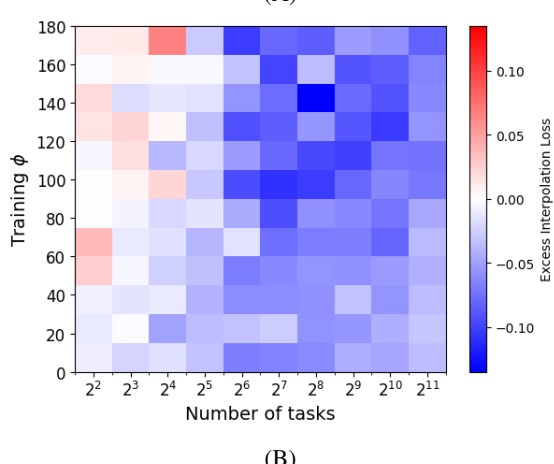

(B)

Figure 14: **In-task-distribution generalization. A:** *Solid:* The loss for a transformer with $\phi = 30°$ along a great circle connecting two weight vectors in the pretraining set. *Dashed:* The loss along the same great circle for the dMMSE estimator (Eqn 10) corresponding to the pretraining set. **B:** Transformers outperform dMMSE for all $\phi$ with sufficiently many pretraining tasks. The color shows the excess loss of the transformer over the dMMSE estimator. When it is negative, the transformer outperforms dMMSE (see Eqn 11).

writing $w = (Z \ y)$ with $Z \in \mathbb{R}$ and $y \in \mathbb{R}^d$, we see that

$$RZ \geq R^2 - r^2/2 \quad (13)$$

We will show that, up to this restriction, $Z = R(2U - 1)$, with $U \sim \text{Beta}(\frac{d}{2}, \frac{d}{2})$. To generate $y$, we first draw $Z$, then draw a vector uniformly from the sphere $S^{d-1}(\sqrt{R^2 - Z^2})$

**Proposition A.1.** $Z = R(2U - 1)$, *with* $U \sim \text{Beta}(\frac{d}{2}, \frac{d}{2})$

*Proof.* Observe that $p(Z)\, dZ$ is proportional to the ($d$-dimensional) surface area of the conical frustum with height $dZ$ and radii $\sqrt{R^2 - Z^2}$, $\sqrt{R^2 - (Z + dZ)^2}$. Expanding

to first order in $\mathrm{d}Z$, the slant height $\ell$ of the frustum is given by:

$$\ell^2 = (\mathrm{d}Z)^2 \left[ \frac{R^2}{R^2 - Z^2} \right] \qquad (14)$$

so that:

$$p(Z)\,\mathrm{d}Z \propto \ell \left( \sqrt{R^2 - Z^2} \right)^{d-1} \qquad (15)$$

$$\propto (R^2 - Z^2)^{(d-2)/2}\,\mathrm{d}Z \qquad (16)$$

Making the change of variables $Z = R(2u - 1)$, we obtain:

$$p(u)\,\mathrm{d}u \propto (u - u^2)^{(d-2)/2}\,\mathrm{d}u \qquad (17)$$

$$\propto u^{\frac{d}{2}-1}(1-u)^{\frac{d}{2}-1}\,\mathrm{d}u \qquad (18)$$

which has the form of a $\mathrm{Beta}(\frac{d}{2}, \frac{d}{2})$ random variable. $\qquad \square$

### A.5.1. SAMPLING $Z$

As we have seen, $Z$ follows a scaled & shifted $\mathrm{Beta}(\frac{d}{2}, \frac{d}{2})$ distribution, except for the constraint that $RZ \geq R^2 - r^2/2$. To implement this constraint, observe first that it is equivalent to:

$$U \geq 1 - \frac{r^2}{4R^2} \qquad (19)$$

Since the distribution of $U \sim \mathrm{Beta}(\frac{d}{2}, \frac{d}{2})$ is symmetric about $1/2$, this condition is equivalent to:

$$U \leq \frac{r^2}{4R^2} \qquad (20)$$

It follows that to sample from $Z$, we can perform the following sequence of transformations:

$$T \sim \mathrm{Unif}\left( 0, F\left( \frac{r^2}{4R^2} \right) \right) \qquad (21)$$

$$U = F^{-1}(T) \qquad (22)$$

$$Z = R(2U - 1) \qquad (23)$$

where $F(\cdot)$ is the cdf of the $\mathrm{Beta}(\frac{d}{2}, \frac{d}{2})$ distribution.

### A.5.2. MINIMUM ANGLES

It is straightforward to implement an additional constraint corresponding to a minimum distance/angle away from the vector $v$. If $r'$ is this minimum distance, then we perform:

$$T \sim \mathrm{Unif}\left( F\left( \frac{(r')^2}{4R^2} \right), F\left( \frac{r^2}{4R^2} \right) \right) \qquad (24)$$

$$U = F^{-1}(T) \qquad (25)$$

$$Z = R(2U - 1) \qquad (26)$$

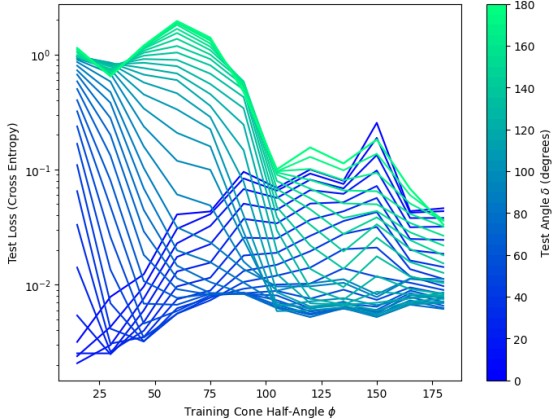

Figure 15: **Data diversity transition:** When instead drawing $x_i$ from a hyperspherical cap, we see a (noisy) specialization-generalization transition at $\phi \approx 105°$.

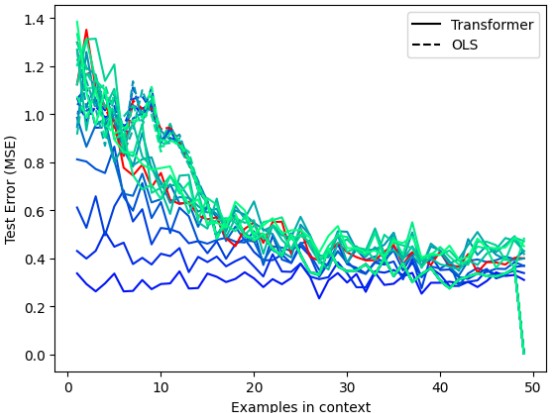

Figure 16: **In-distribution test error vs context length for the noisy case ($\sigma^2 = \frac{1}{4}$).**

### A.6. Data diversity thresholds

In this section we investigate the level of diversity necessary in the *data*: i.e. when we instead draw $x_i \sim S^{d-1}(\phi)$, with $w$ drawn *uniformly* over the *entire* unit sphere. In Fig 15, we show the results. We still observe a transition, suggesting that sufficient diversity is also required in the data in order to generalize out-of-distribution. We notice that the transition is much noisier than that observed in Fig 2. This difference in behavior may be due to the fact that the model directly observes $x_i$ in the context, whereas the task vector $w$ is a latent variable.

### A.7. Different Context Lengths

In Figures 16 and 17, we investigate models trained with various $\phi$ across many context lengths in different settings to the setting in Fig 4.

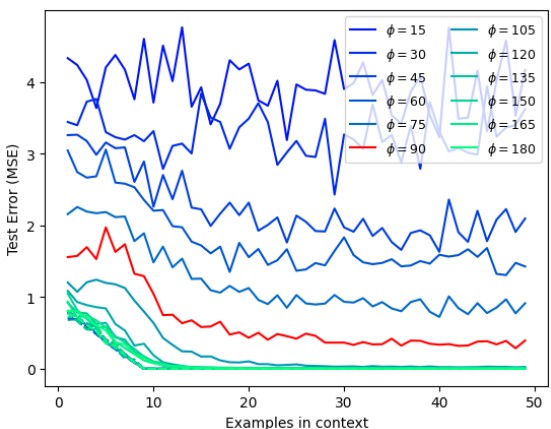

Figure 17: **Out-of-distribution ($\delta = 175°$) test error vs context length.**

**A.8. Supplemental plots for section 3.3**

In Fig 19, we show the data from Fig 9, but for all $\delta$, highlighting that the transition point does not change with increasing model depth.

In Fig 18, we reproduce Fig 10, but show many values of $\delta$, highlighting the non-monotonic behavior that arises as a result of the rotational symmetry of the sphere being violated by the nonlinearity.

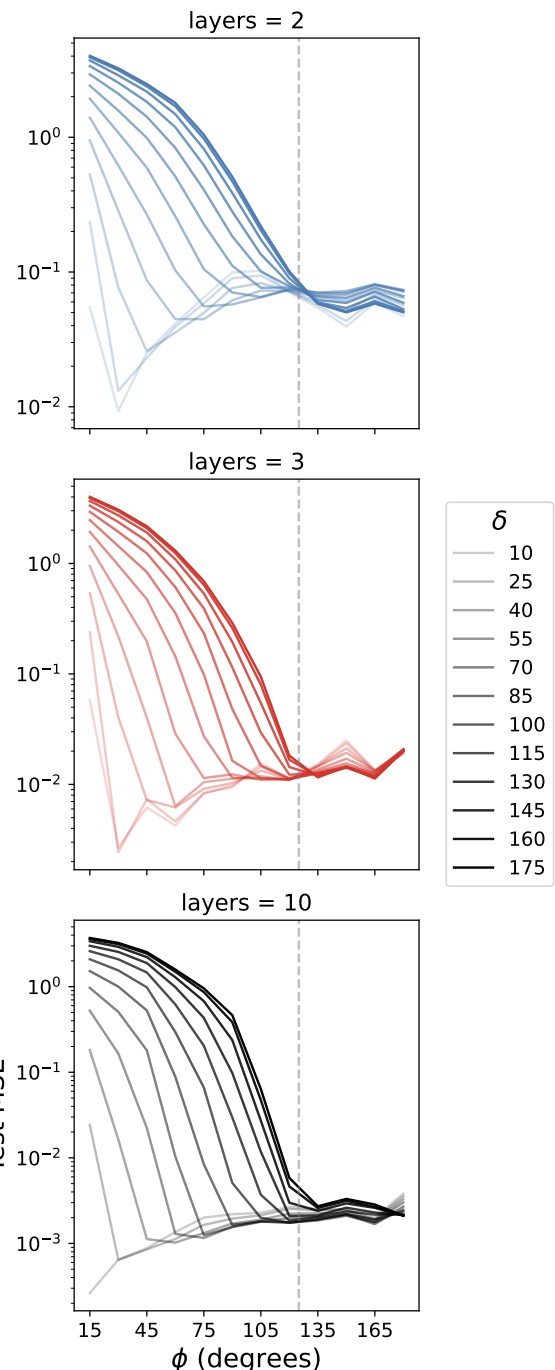

Figure 19: **Distributional diversity threshold is unaffected by model depth, even across test angles.** Test loss vs training spherical cap polar angle $\phi$ from transformer models with two (blue), three (red), and ten (black) layers. We show results for test tasks drawn from $\Delta\delta = 5°$ bands from $\delta = 15°$ to $175°$, where lighter shades indicate smaller $\delta$. We see that the threshold for out-of-task-distribution generalization stays close to $\phi \sim 120°$, regardless of the model depth.

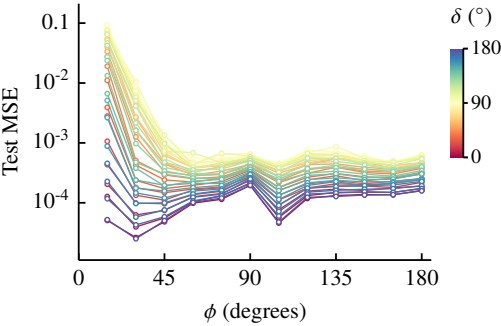

Figure 18: Same as Fig 10A, but shows all values of $\delta$. Notice how the behavior of the test loss is non-monotonic as $\delta$ increases.

