# OpenReview forum: "When can in-context learning generalize out of task distribution?"
_ICML.cc/2025/Conference — ICML 2025 poster_

### Official Review · Reviewer_BMPu · 2025-03-12

**Overall Recommendation:** 3

**Summary:**

This paper examines the generalization properties of in-context learning (ICL) in transformers. Specifically, it explores the conditions necessary for ICL to emerge and extend beyond the pretraining distribution. To investigate this, the authors conduct a series of experiments across various tasks and summarize their key findings.

**Claims And Evidence:**

See strength and weaknesses

**Essential References Not Discussed:**

yes

**Experimental Designs Or Analyses:**

yes

**Methods And Evaluation Criteria:**

yes

**Other Comments Or Suggestions:**

none found

**Other Strengths And Weaknesses:**

## Strengths
- Paper is written clearly and easy to follow
- Paper ask important questions regarding how transformers generalize, a crucial topic for advancing the field’s understanding.

## Weaknesses
- The experiments are relatively simple and do not explore the effects of increasing task complexity. For instance, incorporating vision-language tasks post-pretraining could provide deeper insights into ICL.
- The experimental design appears highly similar to [1], yet the paper does not adequately clarify this resemblance.
- The paper does not clearly answer the key questions posed in the “Contributions” section. While the experimental results section attempts to address them, the discussion lacks clarity and direct conclusions.

1. Raventós, A., Paul, M., Chen, F., & Ganguli, S. (2023). Pretraining task diversity and the emergence of non-bayesian in-context learning for regression. Advances in neural information processing systems, 36, 14228-14246.

**Questions For Authors:**

## Questions to ask
- Can the authors explicitly discuss the similarities between this paper and [1] in terms of experimental design?
- Can the authors provide clearer conclusions drawn from the experimental results regarding the key questions outlined in the “Contributions” section?

1. Raventós, A., Paul, M., Chen, F., & Ganguli, S. (2023). Pretraining task diversity and the emergence of non-bayesian in-context learning for regression. Advances in neural information processing systems, 36, 14228-14246.

**Relation To Broader Scientific Literature:**

NA

**Theoretical Claims:**

yes

---

> ### Author Rebuttal · Authors · 2025-04-01
>
> Thank you for highlighting the clarity and importance of our work! We appreciate your comments and suggestions, which help us improve the paper.
>
> &nbsp;
>
> > The experiments are relatively simple…
>
> We agree that our experimental setups are relatively simple. However, this simplicity is precisely what enables controlled experiments that are essential for developing an understanding of complex learning phenomena. We view our work as establishing an important foundation upon which future research can build more sophisticated models of out-of-task-distribution generalization in increasingly complex settings. This minimal model approach has proved crucial in advancing our understanding of nontrivial learning phenomena, such as generalization and learning dynamics of deep networks (eg, Saxe et al, ICLR 2014; Lampinen & Ganguli, ICLR 2019; Ji & Telgarsky, ICLR 2019; Arora et al, ICLR 2019), double descent and benign overfitting (eg, Bartlett et al, PNAS 2020; Wu & Xu, NeurIPS 2020; Hastie et al, Ann Stat 2022; Richards et al, AISTATS 2021; Mel & Ganguli, ICML 2021), and emergence of in-context learning (eg, Chan & Santoro et al, NeurIPS 2022; Singh & Chan et al, NeurIPS 2023; Raventós et al, NeurIPS 2023; Reddy, ICLR 2024; ​​Nguyen & Reddy, ICLR 2025).
>
> &nbsp;
>
> > …and do not explore the effects of increasing task complexity.
>
> Our work considered several settings with varying task complexity, including noisy regression (Fig 2B), increasing task dimensions (Fig 7), and nonlinear regression (Fig 9). We have also recently performed experiments on a classification task with $y = \mathrm{Heaviside}(w^Tx)$ and $w$ drawn from a hyperspherical cap similar to our linear regression setting. We will include the results in our revised manuscript. In all settings considered, we observe a specialization-generalization transition, similar to those observed in Fig 2A. Our results hint at universal features of out-of-task-distribution generalization in exemplar in-context learning.
>
> We will clarify this point in the discussion section in our revised manuscript.
>
> &nbsp;
>
> > …incorporating vision-language tasks post-pretraining could provide deeper insights into ICL.
>
> We would love to be able to do this, but doing so would require developing a more robust notion of task similarity for vision-language tasks, which is outside the scope of our current work.
>
> &nbsp;
>
> > The experimental design appears highly similar to [Raventós et al], yet the paper does not adequately clarify this resemblance.
>
> > Can the authors explicitly discuss the similarities between this paper and [Raventós et al] in terms of experimental design?
>
> Thank you for pointing out an opportunity to clarify the differences between our work and Raventós et al.
>
> We intentionally chose our experimental setup to begin from Raventós et al. This similarity enables a seamless transition from this earlier work, which considered the effects of the number of training tasks on in-task-distribution generalization, through to our work on out-of-task-distribution generalization. This approach allows for a more direct comparison between previous results and ours, which helps readers build intuitions on the effects of task diversity on the emergence of general purpose in-context learning, without the need to translate between two different experimental settings.
>
> We will revise our manuscript to better explain the rationale behind our experimental design, where we describe our setup.
>
> &nbsp;
>
> > The paper does not clearly answer the key questions posed in the “Contributions” section. While the experimental results section attempts to address them, the discussion lacks clarity and direct conclusions.
>
> > Can the authors provide clearer conclusions drawn from the experimental results regarding the key questions outlined in the “Contributions” section?
>
> We will include a new Conclusions section to the discussion to further clarify how our experiments support our claims in the Contributions section. In particular, we will add the following texts:
>
> * We investigate the ability of transformers trained on simple tasks to generalize out-of-*task*-distribution: where “task generalization” is a particular type of covariate shift.
> * We identify a novel transition in the task generalization performance of transformers trained to do ICL, and show that pretraining tasks need not cover the full task space in order for models to generalize
> * Our experiments identify specialization-generalization transitions in several different ICL problems, suggesting that these transitions may be a universal phenomenon.

---

### Official Review · Reviewer_h9P4 · 2025-03-12

**Overall Recommendation:** 3

**Summary:**

The paper explores the effect of pretraining task diversity, i.e., instead of the number of pretraining tasks, the paper considers the diversity of the fixed number of pretraining tasks.
The same number $N$ of tasks could be more diverse than the other $N$ tasks.
Specifically, the paper draws samples from a subset of a unit hypersphere and tests on the entire hypersphere.
The paper measures the new concept of diversity and shows that there is a transition of the trained Transformer from a specialized solution to generalization over the full tasks when the new diversity of tasks is increased.
There are multiple ablation studies provided to make the experiment complete.

**Claims And Evidence:**

Claims: The paper claims there should be another angle of tasks diversity, i.e., the scale of pretraining task distribution. Via special experimental design, the paper illustrates the claim is true.

Supported? Yes. The claim itself is reasonable and straightforward, and supported by the experiments. The paper could do better via diversifying the experimental setting on the tasks. The paper concentrates on the pretraining task's parameters on the sphere, the paper could consider more either proposing another task, possibly on classification rather than regression, or consider a task distribution other than the sphere.

**Essential References Not Discussed:**

The author could consider to discuss the relationship to:

Can Transformer Models Generalize Via In-Context Learning Beyond Pretraining Data?

**Experimental Designs Or Analyses:**

Yes, I read the full paper so I checked the experimental design. The experiments are straightforward to me.

**Methods And Evaluation Criteria:**

Yes, the experimental setup makes sense.

**Other Comments Or Suggestions:**

There are multiple misuses in the citation. For instance, P1 col2 line038 "the results of (...)". I think the author used \citep, but here \citet should be used since there should be no "()".

P3 col1 line 142 "Fig 10" should be "Fig. 10".

P4 col1 line 201-204 "set by $\sigma^2$. with". should be "With"?

The Optimal Bayes* solution depends on the label noise as shown in equation (5), where $p(w|C,y))$ depends on the label noise over $y$. Therefore, the author may connect the Optimal Bayes* solution to the case of $\sigma\rightarrow 0$.

**Other Strengths And Weaknesses:**

Strength:

(i) Fig 2 illustrates an interesting phenomenon: the pretraining task does not need to cover all the space to achieve good performance on all the space. There is a sweet spoint on the pretraining task distribution which depends on the level of label noise.

Weakness:

(i) The paper should consider arranging some experiments from the appendix into the main paper, especially when the main paper is not full 8 pages and the main paper refers to some figure in the appendix. (P3 col1 line 142 Fig 10 is in the appendix)

The paper is a bit beyond borderline for me, but cannot reach 4, so I score 3. The experiments are good but not surprising, and the claim is very reasonable, well supported, but not surprising. The contribution is decent, well, but not huge. The writing is also good. If the finding could shed some light on how such generalization happens, the paper will score 4.

**Questions For Authors:**

N/A

**Relation To Broader Scientific Literature:**

The key contribution could be regarded as a complement to the definition of pretraining task diversity. There are indeed two angles: the number of training tasks and the scale of pretraining task distribution. The paper highlights the second part. Prior findings are concentrated on the first part.

**Theoretical Claims:**

No theory claim.

---

> ### Author Rebuttal · Authors · 2025-04-01
>
> Thank you for your comments and suggestions! We are pleased that you find our work interesting and well supported.
>
> &nbsp;
> > The paper could do better via diversifying the experimental setting on the tasks.
>
> > …consider more either proposing another task, possibly on classification rather than regression, or consider a task distribution other than the sphere.
>
> Thank you for suggesting an intuitive way to diversify our experimental setting. We have performed experiments on a classification task with binary class labels {0,1}, generated from $y=\mathrm{Heaviside}(w^Tx)$. The tasks $w$ are drawn uniformly from a hypersphere cap similar to our regression setting. We measure model performance on unseen tasks in and out of the training distribution. In this classification setting, we also observe a specialization-generalization transition similar to Fig 2. We update our manuscript to include the new results.
>
> These new results will add to existing variations we considered, such as noisy regression (Fig 2B), increasing task dimensions (Fig 7), and nonlinear regression (Fig 9). A transition between specialized and general purpose solutions was evident in all settings investigated, including the new classification experiments described above. These results suggest that this transition, driven by distributional task diversity, is a universal feature of out-of-task-distribution generalization in exemplar in-context learning.
>
> &nbsp;
> > …consider to discuss the relationship to: [Yadlowsky et al (2023)]
>
> Thank you for pointing out this interesting and relevant work. Yadlowsky et al (2023) asked whether a transformer can in-context learn a novel task and highlight that the coverage of the training task distribution (ie, *distributional task diversity* in our work) is an important factor in ICL ability.
>
> While related, our work differs from theirs in several important aspects:
>
> * Yadlowsky et al’s setting can best be described as “concept shift” of the task distribution, in the context of standard OOD literature. Our work fits within the framework of “covariate shift” of the task distribution.
> * We develop an experimental framework that allows us to clearly define task similarity, which enables quantitative investigations of its effects.
> * We demonstrate that distributional task diversity drives a sharp transition between specialized and general-purpose ICL.
>
> We include discussion of Yadlowsky et al and clarify the points above in our revised manuscript.
>
> &nbsp;
> > …experiments are good but not surprising, and the claim is very reasonable, well supported…
>
> We emphasize that the existence of a specialization-generalization transition in the out-of-task-distribution performance of transformers is nontrivial. One could expect at least three potential outcomes a priori:
>
> 1. No out-of-distribution generalization. This is what one would observe from an optimal Bayesian solution with a prior matching the pretraining distribution.
> 2. Nontrivial out-of-distribution generalization with performance gradually improving with increasing coverage of training task distribution.
> 3. Nontrivial out-of-distribution generalization with abrupt performance improvement as the coverage of training task distribution exceeds a threshold (without encompassing the entire task space).
>
> In our view, the third option, which we found empirically, is perhaps the most interesting. It implies a sharp change in ICL solution patterns where the model learns a new concept that generalizes beyond both the training tasks and the training task distribution.
>
> We discuss these potential options in the introduction of our revised manuscript.
>
> &nbsp;
> > If the finding could shed some light on how such generalization happens, the paper will score 4.
>
> While we don’t have a full theory of the transition to OOD generalization, we suggest that our “covariate shift” of the task distribution setting is critical and helps explain why Yadlowsky et al did not find such a transition. To clarify this, we will add a new section to the discussion that casts the ICL problem as a special case of supervised learning. In this framework, we show how “task generalization” can be seen as a particularly structured type of *covariate shift* or *domain generalization*. In contrast, we show that Yadlowsky et al instead study a type of *concept shift*. We argue that this distinction in the mode of OOD generalization considered is what allows our models to generalize OOD while the models in Yadlowsky et al do not. Our analysis suggests that it may be the type of OOD problem considered, not only the scale of the model/data, that allows for nontrivial OOD generalization in LLMs.
>
> &nbsp;
> > … should consider arranging some experiments from the appendix into the main paper…
>
> We appreciate this helpful suggestion and revise our manuscript accordingly.
>
> &nbsp;
> > There are multiple misuses in the citation…
>
> Thank you for spotting these typos and formatting errors. We fix these in the revision.

---

> > ### Comment · Reviewer_h9P4 · 2025-04-01
> >
> > Thank you for your rebuttal. I have read other reviews and I'll maintain my scores.

---

### Official Review · Reviewer_u8n5 · 2025-03-17

**Overall Recommendation:** 3

**Summary:**

The authors study a new notion of task diversity--task similarity--and investigate the condition on task diversity for out-of-distribution generalization to emerge. The authors find that there is a transition from specialized models to generalizable models with increasing task diversity. They also show that this specialization-generalization transitions also occur in nonlinear regression problems.


## update after rebuttal

I do not have further concerns. I will maintain my score and suggest acceptance of the paper.

**Claims And Evidence:**

I feel that some of the claims may be a little problematic. Specifically, the authors have not discussed how they determine the transition point. For example, in Fig. 8 and 15, it appears from the plot that the threshold slightly increases with number of layers.

**Essential References Not Discussed:**

None

**Experimental Designs Or Analyses:**

I have checked the soundness and validity of experimental designs and analyses. For nonlinear regression experiments, concatenating the weight vectors may cause some unexpected behaviors as the authors have also discussed in section 3.3. Specifically, there is also rescaling symmetry in this nonlinear parameterization. Considering some measure in the function space may make more sense than in the parameter space.

**Methods And Evaluation Criteria:**

The proposed methods and evaluation criteria make sense.

**Other Comments Or Suggestions:**

1.	For Fig. 5B, it may be good to plot normalized MSE loss instead of the test MSE. One can normalize the loss by the variance of y so that the effect of R on the variance of y will be factored out for a clearer point.

**Other Strengths And Weaknesses:**

The paper studies the effect of task diversity on the out-of-distribution performance of in-context learning linear regression from a new perspective. The finding of the paper is interesting, and the paper is overall well written.

However, there are still a few points that the authors can improve.

1.	First, the authors should more clearly write down how they define and find the transition point.

2.	In Fig. 4, it seems that the models with $\phi>120$ will deviate away from OLS for in-context examples more than 10. I suggest the authors adding discussions for this observation.

**Questions For Authors:**

1.	In Fig 5A, why does training on larger $\phi$ have a worse generalization for the cases where test radius is larger than 1?

**Relation To Broader Scientific Literature:**

Understanding the emergence of in-context learning is a very important question. Previous works have shown that the emergence of in-context learning on out-of-distribution linear regression tasks will only occur if the pretraining task is diverse enough. The diversity discussed in previous literatures are about the number of pretraining tasks. Here, the authors take another perspective and view task similarity as another dimension of task diversity. The results of the paper contribute to this broader scientific understanding of in-context learning.

**Theoretical Claims:**

N. A.

---

> ### Author Rebuttal · Authors · 2025-04-01
>
> Thank you for the helpful comments and suggestions!
>
> &nbsp;
>
> > Specifically, the authors have not discussed how they determine the transition point. For example, in Fig. 8 and 15, it appears from the plot that the threshold slightly increases with number of layers.
>
> Thank you for the opportunity to clarify our presentation here. We will introduce a more formal definition of the transition point, which captures our intuition. We say a transition has occurred when the spread of the test error across test angles *δ* = {5°,10°,…} decreases below a specified threshold as the training cone angle *ɸ* increases. We quantify this spread by the relative standard deviation (ie, standard deviation normalized by mean). When plotted against *ɸ*, we generically see a sharp drop in this spread measure as *ɸ* increases.
>
> We include plots of this measure and the thresholded phase boundary in the revised appendix. We will update our presentation, including Figs 8 and 15, to use this more precise definition of the transition.
>
> &nbsp;
>
> > Specifically, there is also rescaling symmetry in this nonlinear parameterization. Considering some measure in the function space may make more sense than in the parameter space.
>
> While we agree that a function space measure would indeed be a more principled approach for nonlinear functions, developing such a measure is beyond the scope of this work. For the nonlinear regression experiments, our main aim is to show that, at least qualitatively, specialization-generalization transitions exist for more complex settings than linear regression. We view the fact that our simple parameter-space measure (which ignores many features important to the nonlinear problem) works at all in this setting as a sign that specialization-generalization transitions may be somewhat robust to a choice of similarity measure.
>
> We will edit the text to clarify this point.
>
> &nbsp;
>
> > Previous works have shown that the emergence of in-context learning on out-of-distribution linear regression tasks…
>
> We would like to emphasize that although previous work has shown the ability of transformers to generalize to linear tasks beyond those they were pretrained on, our work means something different (and, to the best of our knowledge, novel) by “out-of-distribution.” In particular, previous work (eg, Raventos et al) has focused on the ability of transformers to generalize to new tasks *within* the support of the pretraining distribution. Instead, we focus on the ability of transformers to generalize to new tasks which are completely disjoint from those in the pretraining distribution.
>
> In particular, we show in our revised manuscript how “task generalization” can be seen as a particularly structured type of *covariate shift* or *domain generalization* by defining the ICL problem as a special case of supervised learning. Our analysis suggests that it may be the type of OOD problem considered, not only the scale of the model/data, that allows for nontrivial OOD generalization in LLMs.
>
> &nbsp;
>
> > In Fig. 4, it seems that the models with ϕ > 120 will deviate away from OLS for in-context examples more than 10. I suggest the authors adding discussions for this observation.
>
> Thank you for the suggestion to clarify our presentation here. Indeed, in our experiments, models with *ɸ* > 120 do not achieve zero test error after 10 examples. (The test error is just small). While the log scale on the plot exacerbates the difference between these curves, it is possible that this discrepancy is part of what allows transformers to generalize nontrivially out-of-distribution while an optimal Bayesian algorithm (with a prior matching the pretraining distribution) does not.
>
> We will update our manuscript to comment on this phenomenon.
>
> &nbsp;
>
> > For Fig. 5B, it may be good to plot normalized MSE loss instead of the test MSE.
>
> Thank you for the suggestion. If we normalize the MSE accordingly in Fig 5B, the transition remains qualitatively similar. We have added this plot to the appendix (but we leave the current plot in the main text to ease presentation).
>
> &nbsp;
>
> > In Fig 5A, why does training on larger ϕ  have a worse generalization for the cases where test radius is larger than 1?
>
> We believe that this effect is due to initialization/optimization noise in the training of the models. We update the plot with a version that averages over more models to clarify this.

---

> > ### Comment · Reviewer_u8n5 · 2025-04-03
> >
> > I thank the authors for their responses to my questions. And I understand and acknowledge the novelty of the OOD studied in the paper. I do not have further concerns. I will maintain my score and suggest acceptance of the paper.

---

### Official Review · Reviewer_STGS · 2025-03-18

**Overall Recommendation:** 3

**Summary:**

This paper studies empirically the task diversity to the generalizability, focusing on the transformer trained to learn a linear regression problem. It proposes the new axis of task diversity, namely the task similarity, independent of the unique task numbers seen during pretraining.

**Claims And Evidence:**

See experimental section for questions.

**Essential References Not Discussed:**

Not that I aware of.

**Experimental Designs Or Analyses:**

1. Could you explain why you mention the "loss plateaus" behavior in section 3? Are the results presented with the runs that "escape" the plateaus?
2. Could you explain why the noise level sigma affects the transition point? How to filter out the effect that because of the non-zero noise level, the "effective training degree" is larger than the set training degree?
3. In Figure 2, when \delta is small, the test MSE is better for small \phi than large \phi. I assume this has to do with the fact that the test MSE for a shortcut solution trained for small \phi is small, instead of transformer trained on small \phi learns a better solver. Is there any way to normalize the test MSE such that we will not observe the confusing behavior where an increase in test MSE when \phi increases?  Maybe use the way you normalize the loss in section 3.2?
4. How do you plot figure 6C? What is the threshold chosen, and why? How do you validate the IWL regime?
5. Could you properly define the "transition" point? Is it the \phi that across \delta the test MSE collapses? Or is it the \phi that the test MSE plateaus when \delta=175? In Figure 2 I assume the definition goes by the first, and in figure 7 the definition goes by the second. Are these two definitions always collapses?
6. In Figure 7, what is the context length? What training algorithm did you use to train especially the task with large d? Are all the runs have saturated performance (that escape the loss plateaus)?
7. Out of the whole results presented in the paper, I am mostly interested in the results in Figure 7 & 8. Could there be any explanation on why the transition happens independent of the problem dimension and model depth? Could it have to do with model embedding size?

**Methods And Evaluation Criteria:**

Yes

**Other Comments Or Suggestions:**

- radii in line 272

**Other Strengths And Weaknesses:**

I appreciate the paper’s effort in introducing a new perspective on task diversity measurement and providing extensive experimental evidence to empirically analyze model behavior. The work is well-motivated and contributes valuable insights into understanding task generalization.

That said, the transition from an exemplar-based solver to a more generalizable solver is a well-studied phenomenon, so some of the initial findings may not be particularly surprising. For example, Figure 2 illustrates the existence of a transition point, while Figures 3 and 4 highlight that transformers learn to solve tasks using a different prior than OLS. Figure 5 further shows generalization beyond the unit sphere, where models trained with small \phi struggle with OOD tasks within the unit sphere. Lastly, Figure 6 provides a helpful visualization of the phase transition in terms of task numbers and similarity. While these results are interesting and support the overall argument, they largely align with existing expectations.

**Questions For Authors:**

Please see the experimental section. Thanks!

**Relation To Broader Scientific Literature:**

This paper expand on the previous work on the effect of task diversity in generalizability of the transformer.

**Theoretical Claims:**

NA

---

> ### Author Rebuttal · Authors · 2025-04-01
>
> Thank you for your helpful comments and suggestions!
>
> &nbsp;
> > why [do we] mention the "loss plateaus"?
>
> We mention the loss plateaus because they are characteristic of ICL behavior (Fu et al, ICML 2024; Reddy, ICLR 2023), confirming that our training is consistent with ICL phenomenology.
>
> We will move this sentence to a footnote.
>
> &nbsp;
> > Are … [these] runs that "escape" the plateaus?
>
> Yes, all our experiments escape the plateau.
>
> &nbsp;
> > How to [understand that], the "effective training degree" is larger than the set training degree?
>
> With non-zero label noise *σ*, the effective training degree does not change. The training degree, *ɸ*, controls the portion of the hypersphere that the tasks, *w*, are drawn from. The tasks do not change when noise is added as the noise level only affects the label, *y*. Specifically, the noise, *ε* ~ 𝒩(0,*σ*²), is added to *w·x*.
>
> &nbsp;
> > Could you explain why the noise level sigma affects the transition point?
>
> We posit that because of the noisy information that the model sees during training, it takes a larger proportion of the hypersphere to learn a general solution to the linear regression task, moving the transition slightly.
>
> &nbsp;
> > In Figure 2, when \delta is small, the test MSE is better for small \phi than large \phi. … [Does the model learn a shortcut solution?]
>
> Yes, this “shortcut” solution is the specialized solution the transformer develops when only training on small *ɸ*. The transformer has not seen enough of the hypersphere in training to generalize to large test angles (*δ*) it has never seen before.
>
> &nbsp;
> > Is there any way to normalize the test MSE …?
>
> We have made a plot with normalized test MSE to disentangle any effects from unnormalized test MSE and to instead focus on the transition from specialization at small *ɸ* to generalization at large *ɸ*. Here, the test MSE starts at the same value and avoids this behavior of the unnormalized test MSE being higher at small test angles *δ* when trained on large *ɸ*. However, this does not change the transition to generalization, illustrated in Fig 2.
>
> &nbsp;
> > How do you plot figure 6C? What is the threshold chosen, and why?
>
> Currently, the information is only in the figure caption—Thank you for reminding us to add this important detail to the main text. The threshold is set to 0.01 where losses below this threshold for the in-distribution (ID) and out-of-distribution (OOD) losses are characterized as out of task distribution generalization (purple), losses below the threshold for the ID losses but above this threshold for OOD losses are characterized as in task distribution generalization (yellow), and losses above this threshold for both ID and OOD losses are characterized as IWL (teal).
>
> &nbsp;
> > How do you validate the IWL regime?
>
> We validate the IWL regime in appendix section A.3. We note that this is the same methodology used by Raventos et al.
>
> &nbsp;
> > Could you properly define the "transition" point?
>
> We have produced new plots clarifying how we define the transition point that we add to the text. Here, we define the transition point as the reduction in the standard deviation of the mean over *δ*, leading to the “collapse” in the MSE. In Figs 7 and 8, we only plot the results for *δ* = 175°. Results for all values of *δ* for Fig 8 are shown in Fig 15 in the appendix, which emphasizes that the transition is the same regardless of the value of *δ* plotted, where we see the collapse of the MSE (similar to the results in Fig 2).
>
> &nbsp;
> > In Figure 7, what is the context length? What training algorithm did you use …
>
> We use the same context length (*n*=50 examples) and training algorithm (AdamW) across all runs, regardless of task dimension, *d*.
>
> &nbsp;
> > Are all the runs have saturated performance …?
>
> The loss converges for all runs. The loss increases across task dimensions because the regression problem becomes more difficult for the transformer to solve. However, we still see the same transition point in the training angle *ɸ* where the transformer begins to learn a general solution.
>
> &nbsp;
> > [Explain] why the transition happens independent of the problem dimension and model depth? Could it have to do with model embedding size?
>
> The model embedding size remains the same for all experiments. We believe that the transition occurs independently of the linear regression task dimension (as in Fig 7) and the number of layers of the transformer (as in Figs 8 and 15) because of the structure in the data itself. The similarity in the data, independent of the dimension of the hypersphere, is enough for the model to learn a general solution that extends to the entire hypersphere after seeing tasks drawn from only *ɸ*=120°. While previous work, such as Raventos et al (2023), focuses on the number of tasks needed to generalize, our experiments suggest that this task similarity measure is also crucial in moderating the learning of a generalized solution.

---

> > ### Comment · Reviewer_STGS · 2025-04-05
> >
> > I appreciate the author’s response. I have carefully read it and decided to maintain my score.
> >
> > Reason for not a higher score: While it is interesting to explore a new axis of task diversity, the current setup for determining the degree of transition still feels somewhat hand-wavy. It primarily relies on interpreting the test MSE curve. Providing a more rigorous and formal definition of the transition would strengthen the work.
> >
> > Reason for not a lower score: The paper conducts extensive analyses along this new axis of task diversity and presents several insightful experiments.

---

> > > ### Author Response · Authors · 2025-04-09
> > >
> > > Thank you for your response; we appreciate that you find our experiments insightful!
> > >
> > > &nbsp;
> > >
> > > We agree that our approach to determining the transition angle involves heuristics. However, we would like to offer the following points of clarification.
> > >
> > > First, developing a rigorous definition of this specialization-generalization transition would require developing an analytic theory, which is outside the scope of our current experimental work. Sharp phase transitions in statistical physics typically occur only in the idealized thermodynamic limit (i.e., infinite systems). In finite systems, it is nontrivial to unambiguously pinpoint the transition point as it becomes blurred quite generally.
> > >
> > > Second, our “experiment first” approach parallels the natural progression in this research area. Raventos et al (NeurIPS 2023) empirically identified the in-distribution axis of task diversity without providing a formal definition of the transition point or developing a corresponding theory. Only later did Lu et al (M3L Workshop @ NeurIPS 2024) develop such a theoretical framework with formal definitions. Our extensive experiments provide substantial empirical evidence for a specialization-generalization transition in the out-of-distribution generalization behavior of transformers, which also provides fertile ground for subsequent theory.
> > >
> > > &nbsp;
> > >
> > > [1] Pretraining task diversity and the emergence of non-Bayesian in-context learning for regression, Allan Raventós, Mansheej Paul, Feng Chen, Surya Ganguli, NeurIPS 2023
> > >
> > > [2] Yue M. Lu, Mary I. Letey, Jacob A. Zavatone-Veth, Anindita Maiti, Cengiz Pehlevan, Asymptotic theory of in-context learning by linear attention, NeurIPS M3L Workshop 2024

---

### Decision · Program_Chairs · 2025-05-01

**Decision:**

Accept (poster)

**Comment:**

In this work, the authors propose a framework for probing the emergence of in-context learning in transformers.  They focus on the linear regression setting where each task (regression vector) has angle at most $\phi$ with another fixed vector.  By changing this angle from pretraining and test time, this allows for a nice, quantitative measure to probe whether a good algorithm is learned based on how diverse your pretraining data is.  The authors find that there are sharp transitions in the ability for models to generalize to arbitrary data based on the diversity of the pretraining data, as well as a number of other interesting results.  The reviewers were largely in favor of the work, and I'm inclined to agree, and recommend publication.